# Alcohol Deprivation Differentially Changes Alcohol Intake in Female and Male Rats Depending on Early-Life Stressful Experience

Marielly Carvalho [1,2,†], Gessynger Morais-Silva [3,4,†] , Graziele Alícia Batista Caixeta [1,2], Marcelo T. Marin [3,4] and Vanessa C. S. Amaral [1,2,*]

1    Laboratory of Pharmacology and Toxicology of Natural and Synthetic Products, State University of Goias, Exact and Technological Sciences Campus, Anapolis 75132-903, CO, Brazil; marielly.carvalho.mc@gmail.com (M.C.); grazielealicia2015@gmail.com (G.A.B.C.)
2    Graduate Program in Sciences Applied to Health Products (PPGCAPS) UEG, Anápolis 75132-903, GO, Brazil
3    Laboratory of Pharmacology, School of Pharmaceutical Sciences, Sao Paulo State University (UNESP), Araraquara 14800-903, SP, Brazil; gessynger.morais@unesp.br (G.M.-S.); marcelo.marin@unesp.br (M.T.M.)
4    Joint Graduate Program in Physiological Sciences (PIPGCF) UFSCar/UNESP, São Carlos, Araraquara 14801-903, SP, Brazil
*    Correspondence: vanessa.cristiane@ueg.br
†    These authors contributed equally to this work.

**Abstract:** Experiencing early-life adverse events has enduring effects on individual vulnerability to alcohol abuse and the development of addiction-related behaviors. In rodents, it can be studied using maternal separation (MS) stress. Studies have shown that, depending on the protocol used, MS can affect the mother and pups' behavior and are associated with behavioral alterations later in adulthood, associated with both positive or negative outcomes. However, it is not fully elucidated how MS affects relapse-like behaviors when experienced by female or male individuals. Therefore, the aim of our study was to evaluate the effects of brief and prolonged MS on the alcohol deprivation effect (ADE) in female and male rats. Female and male Wistar rats were exposed to brief (15 min/day) or prolonged (180 min/day) MS from postnatal day (PND) 2 to 10. Later, during adulthood (PND 70), animals were submitted to an ADE protocol. Brief MS exposure prevented the ADE in both females and males, while prolonged MS exposure also prevented the ADE in female rats. Moreover, the ADE was more robust in females when compared to males. In conclusion, we showed that male and female rats are differentially affected by alcohol deprivation periods depending on their early-life experiences.

**Keywords:** addiction; maternal separation; risk factor; sex differences

## 1. Introduction

The main feature involved in alcohol addiction is the neuroplasticity that undergoes the neurocircuitry related to motivation, habit, learning, and cognitive control. These neuroplastic events occur due to chronic drug intake and involve complex interactions with the environment and genetic factors [1]. Such long-lasting neuronal alterations are a crucial part of the most challenging step related to the treatment of addiction, namely relapse [2].

Given the pivotal role of neuroplasticity in the development of addiction, it is not surprising that early-life adverse events have enduring effects on individual vulnerability to alcohol abuse and the development of addiction-related behaviors [3]. A history of childhood maltreatment increases the incidence of alcohol use disorder (AUD) in adolescents [4] and adults [5,6] and is related to an earlier onset of drinking and alcohol abuse [7,8] and persistence of alcohol-related disorders through life [9]. Moreover, the greater the severity of the childhood abuse experienced, the greater is the impact on the prevalence and severity of the psychiatric disorders later in life [6,10]. Some studies highlight that such effects could be sex-dependent since they are more prevalent in women [11,12]. Female and male

rats exposed to early life stress show an early maturation of the connections between the basolateral amygdala and the medial prefrontal cortex, while this alteration occurs even earlier in females compared to males [13]. Female rats exposed to early life adversity also show an increase in serotonin metabolites in the ventral tegmental area compared to males [14]. Frontal cortex maturation occurs earlier in females than males [15].

Sex differences in alcohol drinking behavior and the incidence of AUD are extensively reported in the literature. In rodents, most studies show that ethanol intake in females is greater than males' [16–20], while men drink more and have a greater prevalence of AUD [21]. Although such data seems to be contradictory to rodent studies, human drinking behavior is largely influenced by social factors [22,23]. Moreover, recent reports are showing that women are drinking more, facing more problems with alcohol drinking, and women who drink excessively are more susceptible to the development of AUD than men [24].

The alcohol deprivation effect (ADE) is a broadly used rodent paradigm to study relapse-like drinking and has been exploited to the screening of possible candidates for the treatment of AUD relapse [25–27]. It consists of the chronic exposition to voluntary ethanol intake followed by periods of ethanol deprivation and re-exposure, resulting in a temporary increase in intake and preference [25]. The FDA (USA Food and Drug Administration) approved medications naltrexone and acamprosate, block the ADE in rodents, and are used for the treatment of AUD in humans [28–30].

Maternal separation (MS) is an animal model widely used to study the effects of early-life stress on alcohol intake and abuse. It relies on the importance of the pups-mother relationship, especially during the first two weeks (at least in rodent models) and, can be roughly separated into two main categories: the brief and prolonged MS [31]. These conditions affect the mother and pups' behavior and are associated with behavioral alterations later in adulthood, associated with both positive and negative outcomes, depending on the protocol used [31–33]. In the brief MS, the pups are kept apart from their dams for short time intervals (usually 3 to 15 min), which mimics the natural environment of these animals (in situations such as dams leaving the nest to go after food, for example). In prolonged MS, the pups are separated from their dams for prolonged time intervals (from 60 min to 24 h) [34]. While the brief MS has been associated with positive behavioral outcomes, reducing the effects of stress exposure later in life [33], the prolonged MS has been associated with negative outcomes, increasing the stress reactivity later in life [35].

In animal models, a history of early-life stress also seems to impact the individual vulnerability to addiction-related behaviors and alcohol abuse. As well as for stress responses, such effects are highly dependent on the protocol used. Most of the studies have shown that the brief MS seems to decrease while the prolonged MS increases ethanol intake in rats when the animals are tested in adult life [16,31,36–38]. Prolonged MS also increases the impulsivity for alcohol consumption in adult rats [39] and the ethanol conditioned place preference (CPP) acquisition in adolescent rats, while it does not impact CPP in adult rats [16]. However, there are no studies examining MS effects on animal models of ethanol intake relapse. Thus, considering the importance of the relapse behavior to understanding ethanol addiction and to the development of treatment strategies, we studied the effects of brief and prolonged MS on the alcohol deprivation effect, a rodent model for the study of ethanol intake relapse [25] in female and male rats.

## 2. Materials and Methods

### 2.1. Animals

Female and male Wistar rats from the State University of Goias were used for mating and as experimental animals. Experiments were carried out according to the principles and standards of the National Council for the Control of Animal Experimentation (CONCEA), based on NIH Guidelines for the Care and Use of Laboratory Animals as approved by the Commission on Ethics in Animal Use (CEUA) of the State University of Goias (protocol number 008/2016).

Forty-five adult Wistar rats (females: n = 30; males: n = 15) were used for mating. Primiparous females were mated with adult males in a 1:1 ratio, and gestation day 0 (GD0) was defined as the day when copulatory plugs and/or sperm were found. Pregnant rats were kept alone during gestation and checked daily for newborn pups.

Seventy-six Wistar rats (males: n = 39; females: n = 37) were used as experimental animals. After weaning (postnatal day 21 - PND 21), they were kept in groups of 3–5 animals/cage, until the beginning of the alcohol deprivation effect experiment (at PND 70), when they were individually housed. The females' estrous cycle was not pharmacologically synchronized.

Food and water were available *ad libitum* through the experiments. Animals were maintained in polypropylene cages (41 × 34 × 16 cm), under a 12 h light/dark cycle in a temperature-controlled environment (22 ± 2 °C).

### 2.2. Maternal Separation (MS)

The brief and prolonged MS was performed as previously described [16]. From PND 2 to 10, pups were daily separated from their dams for 15 minutes (brief MS – MS15) or 180 min (prolonged MS – MS180). During the separation period, the dams were transferred to clean cages and moved to a separate room, away from the pups. The pups were kept in their nests, under a red incandescent light, maintaining the temperature at 32 ± 2 °C. Weaning was normally performed at PND 21. The control group was kept undisturbed with the dams (except for cage cleaning) until weaning. To avoid any bias related to the litter, a maximum of two pups of each sex, from each dam, were used in each experimental group.

### 2.3. Alcohol Deprivation Effect (ADE)

The ADE protocol was adapted in our laboratory from the previously described [40].

Starting on PND 70, animals were exposed, for two days, to two bottles containing ethanol (6%) as the only source of fluid. Next, animals were given free access to two bottles, one bottle containing crescent ethanol concentrations (6, 8, and 10%) and one containing filtered water, for 4 days each concentration. This phase was called habituation and was intended to initiate alcohol drinking and habituate animals to the taste of the ethanol and experimental procedure. The ethanol consumption of each concentration was analyzed separately and identified as follows: ETOH 6% phase, ETOH 8% phase, and ETOH 10% phase. After that, baseline consumption was recorded: animals were given free access to two bottles, one containing ethanol (10%) and one containing filtered water for 10 days. Baseline consumption was considered as the average of the 4 days before the beginning of the deprivation phase (days in both male and female control groups with less than 20% variation from the average intake of ethanol 10%).

The next day after the baseline consumption phase, animals were submitted to the alcohol deprivation phase. Each deprivation period was comprised of 14 days of alcohol deprivation when only one bottle containing water was available followed by 7 days of free access to one bottle of 10% ethanol solution and one containing filtered water. The ADE was analyzed using the average consumption of four days after the end of each deprivation period. The animals were submitted to two deprivation periods. The experimental procedure indicating each ethanol exposure and deprivation phase is depicted in Figure 1.

Ethanol solutions (*v/v*) were prepared from absolute ethanol (Neon Comercial, São Paulo, SP, Brazil) diluted in filtered water and offered in plastic bottles fitted with stainless steel sipper tubes with ball-valve nipples in rubber stoppers. Ethanol and fluid intake was measured by weighing the bottles and reported as relative to body weight (g/kg). Liquid loss by leakage or evaporation was accessed using bottles placed in empty cages and was subtracted from the amount consumed by each animal.

Solutions were prepared fresh every day and presented to the animals at the same time. The position of the bottles was alternated every day to avoid side preference.

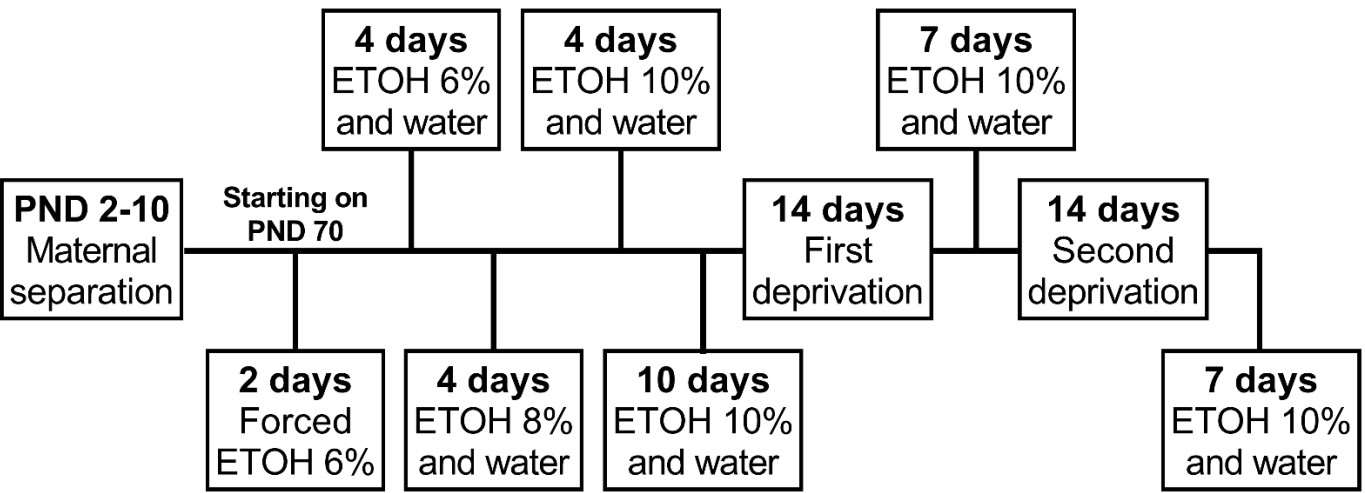

**Figure 1.** Experimental procedure timeline. Female and male rats were exposed to brief or prolonged maternal separation during early life and submitted to an alcohol deprivation effect protocol during adulthood. PND, postnatal day; ETOH, ethanol.

*2.4. Statistics*

Values were expressed as mean ± SEM and analyzed by repeated-measures ANOVA considering the independent factors sex and MS and the repeated measure phases. When ANOVA showed significant differences ($p \leq 0.05$), the Newman-Keuls *post hoc* test was performed.

Statistics were performed using Statistica 7.1 software (StatSoft, Inc., Tulsa, OK, USA) and graphs constructed using GraphPad Prism 7 software (GraphPad Software Inc., La Jolla, CA, USA).

## 3. Results

*3.1. Ethanol Intake Prior to Deprivations*

We found a significant difference in ethanol intake before deprivations, related to the sex and the phase of the experiment, but not MS (Figure 2). The repeated-measures ANOVA showed a significant effect for the factor sex ($F_{1,69} = 56.07$; $p < 0.001$), phase ($F_{3,207} = 13.14$; $p < 0.001$), and the interaction between sex and phase ($F_{3,207} = 3.50$; $p < 0.05$). The Newman-Keuls test revealed that females consumed more ethanol than males in all phases of the experiment before deprivations ($p < 0.001$). In males, ethanol intake during the baseline phase was increased when compared to ethanol consumption in ETOH 6% phase ($p < 0.001$). In females, ethanol consumption during the ETOH 10% phase was the greater prior deprivations ($p < 0.05$), while baseline ethanol intake was increased when compared to ethanol intake during ETOH 6% ($p < 0.01$) and ETOH 8% phases ($p < 0.05$).

*3.2. Alcohol Deprivation Effect*

Alcohol deprivation differentially affected the ethanol consumption in males and females, depending on the early-life experiences (brief or prolonged MS) (Figure 3A and B). The repeated-measures ANOVA showed a significant effect for MS ($F_{2,69} = 4.52$; $p < 0.05$), sex ($F_{1,69} = 17.59$; $p < 0.001$), phase ($F_{2,138} = 29.07$; $p < 0.001$), and for the interaction between MS and phase ($F_{4,138} = 3.22$; $p < 0.05$) and, MS, sex and phase ($F_{4,138} = 4.06$; $p < 0.01$).

Control males showed an increase in ethanol consumption only after the second deprivation ($p < 0.05$) compared to baseline. MS180 male group, similar to control males, showed a significant increase in ethanol consumption only after the second deprivation (emphp < 0.001) compared to baseline intake, while the MS15 male group did not show significant alterations in ethanol consumption after neither the first or second deprivation phases relative to baseline (Figure 3A).

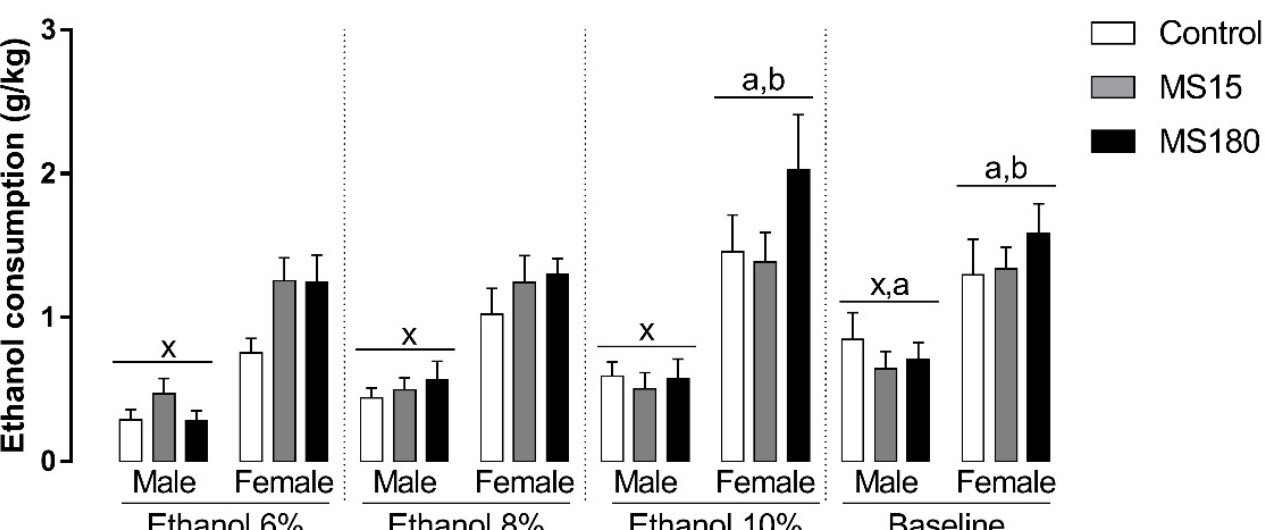

**Figure 2.** Ethanol intake before deprivations in female and male rats exposed to brief and prolonged maternal separation. Bars represent the mean ± SEM (n = 11–14 animals/group). Each concentration represents the average intake of the 4 days of exposition in each experimental phase of the procedure. x; $p < 0.05$ when compared to female groups of the same experimental phase, independent of manipulation; a, $p < 0.05$ when compared to the ETOH 6% phase of the same sex, independent of manipulation; b, $p < 0.05$ when compared to ETOH 8% phase of the same sex, independent of manipulation; c, $p < 0.05$ when compared to ETOH 10% phase of the same sex, independent of manipulation. ETOH, ethanol; MS15, brief maternal separation; MS180, prolonged maternal separation.

Control females showed an increase in ethanol consumption after either the first ($p < 0.01$) or second deprivation ($p < 0.001$) compared to baseline. Ethanol intake after the second deprivation was also greater than ethanol intake after the first deprivation ($p < 0.01$). Both MS15 and MS180 female groups did not show any significant alterations in ethanol consumption after neither the first or second deprivation phases. Additionally, ethanol consumption of the control female group was greater than the MS15 female group ($p < 0.05$), MS180 female group ($p < 0.05$) and control male group ($p < 0.05$) after the second deprivation (Figure 3B).

We also found significant alterations in ethanol preference (Figure 3C,D). The repeated-measures ANOVA showed a significant effect for MS ($F_{2,69} = 3.70$; $p < 0.05$), phase ($F_{2,138} = 37.68$; $p < 0.001$), and for the interaction between sex and phase ($F_{2,138} = 4.06$; $p < 0.05$) and, MS, sex and phase ($F_{4,138} = 2.41$; $p < 0.01$).

Control males showed an increase in ethanol preference only after the second deprivation compared to baseline ($p < 0.001$) and after the first deprivation ($p < 0.05$). MS180 male group, similar to control males, showed a significant increase in ethanol preference only after the second deprivation ($p > 0.001$) compared to baseline and after first deprivation ($p < 0.05$), while the MS15 male group did not show significant alterations in ethanol preference after neither the first or second deprivation phases relative to baseline (Figure 3C). Similar to control males, control females showed an increase in ethanol preference after the second deprivation compared to baseline ($p < 0.001$) and after the first deprivation ($p < 0.01$). MS15 and MS180 female groups did not show an increase in ethanol preference after neither the first or second deprivation phases. Ethanol preference of the control female group was greater than the MS180 female group ($p < 0.05$) after the second deprivation (Figure 3D).

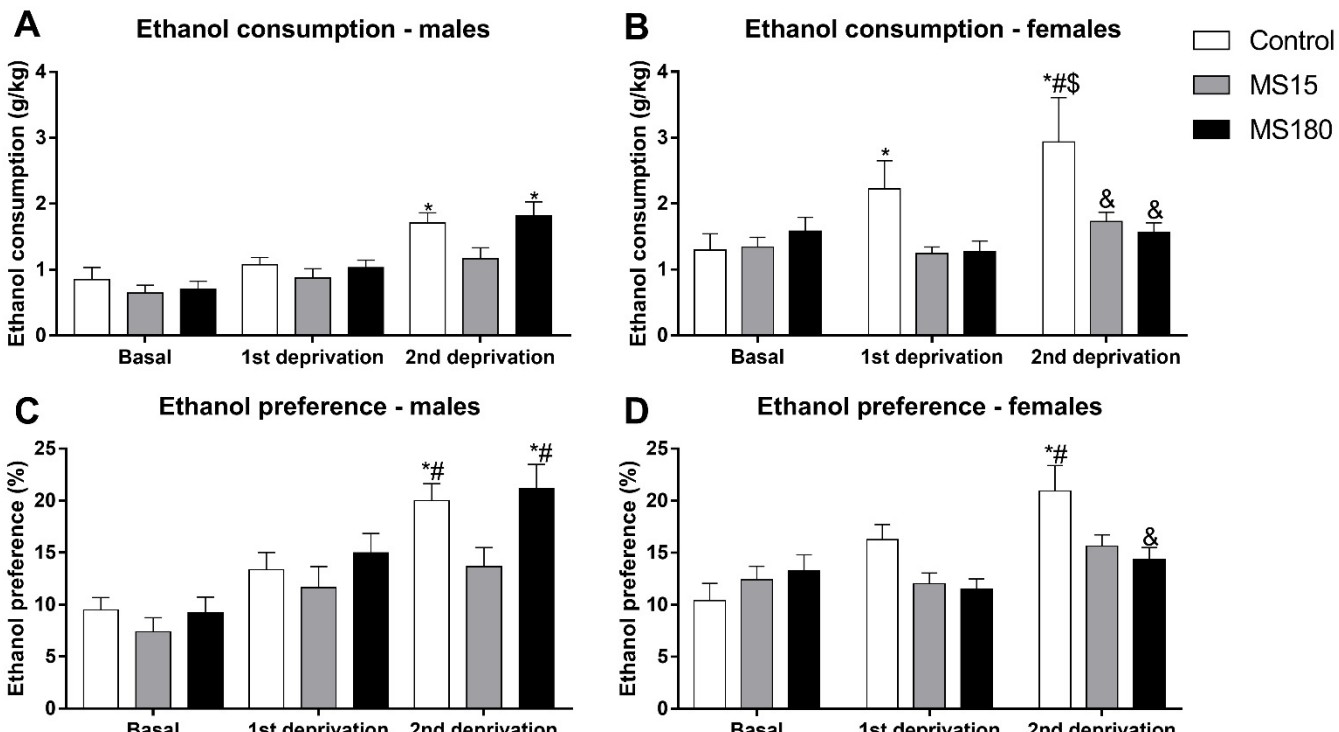

**Figure 3.** Alcohol deprivation effect (ADE) in female and male rats exposed to brief or prolonged maternal separation. The ADE was analyzed using the average ethanol intake or preference of four days after the end of each deprivation period compared to the last four days of the baseline period. The animals were submitted to two deprivation periods. Bars represent the mean $\pm$ SEM (n = 11–14 animals/group). (**A**), male ethanol consumption (g/kg); (**B**), female ethanol consumption (g/kg); (**C**), male ethanol preference; (**D**), female ethanol preference. *, $p < 0.05$ relative to the baseline ethanol consumption of the same gender control group; #, $p < 0.05$ relative to the ethanol consumption after the first deprivation of the same gender control group; &, $p < 0.05$ relative to the ethanol consumption after the second deprivation of the same gender control group; \$, $p < 0.05$ relative to the ethanol consumption of the control male group after the second deprivation. MS15, brief maternal separation; MS180, prolonged maternal separation.

### 3.3. Fluid Intake

There was a significant effect of sex and phase, independent of the MS, on fluid intake through the experiment (Figure 4). The repeated measures ANOVA showed a significant effect for sex ($F_{1,69} = 201.83$; $p < 0.001$), phase ($F_{5,345} = 30.61$; $p < 0.001$) and for the interaction between sex and phase ($F_{5,345} = 2.60$; $p < 0.05$). Overall, female fluid intake was increased compared to males ($p < 0.001$). Moreover, fluid intake, independent of the sex or MS decreased through the experimental phases. In males, fluid intake in ETOH 10%, baseline, 1st re-exposure, and 2nd re-exposure phases was smaller than fluid intake in ETOH 6% and ETOH 8% phases, while fluid intake during 1st re-exposure was smaller than ETOH 10% phase. In females, baseline, 1st re-exposure, and 2nd re-exposure fluid intake were smaller than ETOH 6%, ETOH 8%, and ETOH 10% phases. The female fluid intake in ETOH 10% phase was smaller than fluid intake in ETOH 6% phase.

### 3.4. Body Weight

Body weight was evaluated through the different phases of the ADE protocol and reported as body weight gain in each period (Figure 5). First, we found a significant effect in initial body weight (Figure 5A) for the factor maternal separation ($F2,69 = 5.06$; $p < 0.01$) and sex ($F1,69 = 567.40$; $p < 0.001$). The post-hoc analysis revealed that males began the alcohol deprivation effect protocol heavier than females ($p < 0.001$), and both brief or prolonged maternal separation increased body weight in both males and females ($p < 0.05$).

On the other hand, total body weight (Figure 5B) through the experiment was only affected by sex ($F_{1,69}$ = 55.57; $p < 0.001$), revealing that males gain more weight than females over time ($p < 0.001$). Next, we evaluated the body weight gain in each phase of our protocol (Figure 5C). Body weight gain during baseline phase was affected only by sex ($F_{1,69}$ = 4.21; $p < 0.05$), revealing a greater body weight gain in males compared to females ($p < 0.05$). During the first deprivation period, the two-way ANOVA revealed a significant effect for maternal separation ($F_{2,69}$ = 5.51; $p < 0.01$) and sex ($F_{1,69}$ = 35.44; $p < 0.001$), showing that males gained more weight during this phase ($p < 0.001$) and maternal separation increased body weight gain during the ethanol deprivation phase ($p < 0.05$). We found similar results for the first re-exposure phase: a significant effect for maternal separation ($F_{2,69}$ = 10.88; $p < 0.01$) and sex ($F_{1,69}$ = 7.19; $p < 0.001$), showing that males gained more weight during this phase ($p < 0.001$) and maternal separation increased body weight gain during the ethanol deprivation phase ($p < 0.001$). We found a significant effect only for sex during the second deprivation phase ($F_{1,69}$ = 7.90; $p < 0.01$), when males gained more weight than females ($p < 0.01$). Finally, we found a significant effect for the interaction between maternal separation and sex ($F_{2,69}$ = 7.63; $p < 0.01$) in the second re-exposure phase, while the post-hoc test did not reveal any significant differences.

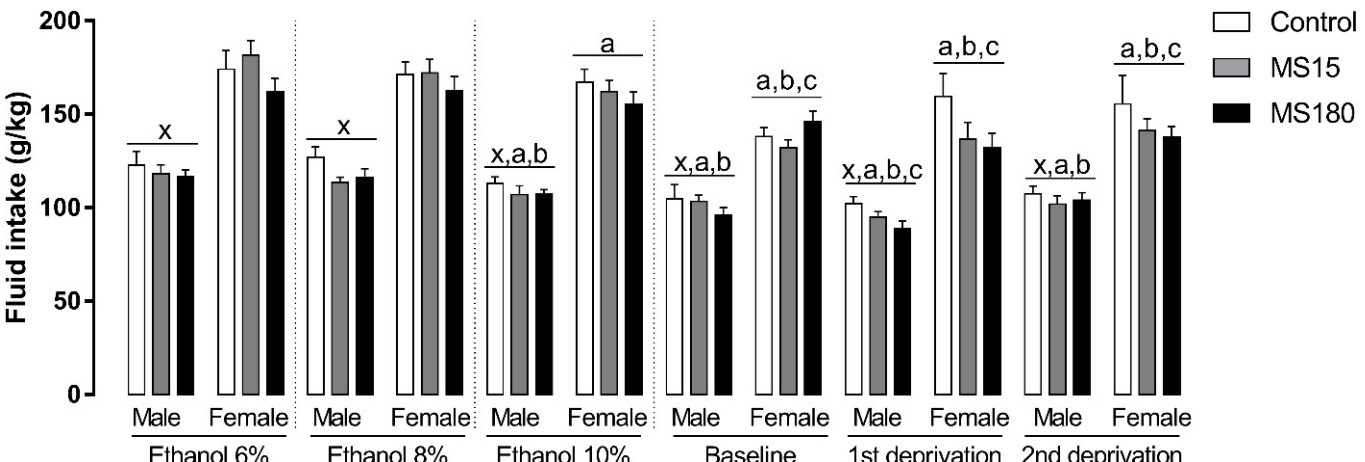

**Figure 4.** Fluid intake through the experiment in female and male rats exposed to brief and prolonged maternal separation. Bars represent the mean ± SEM (n = 11–14 animals/group). Each concentration represents the average intake of the 4 days of exposition in each experimental phase of the procedure. x; $p < 0.05$ when compared to female groups, independent of the manipulation or experimental phase; a, $p < 0.05$ when compared to the ETOH 6% phase, independent of sex or manipulation; b, $p < 0.05$ when compared to ETOH 8% phase, independent of sex or manipulation; c, $p < 0.05$ when compared to ETOH 10% phase, independent of sex or manipulation. ETOH, ethanol; MS15, brief maternal separation; MS180, prolonged maternal separation.

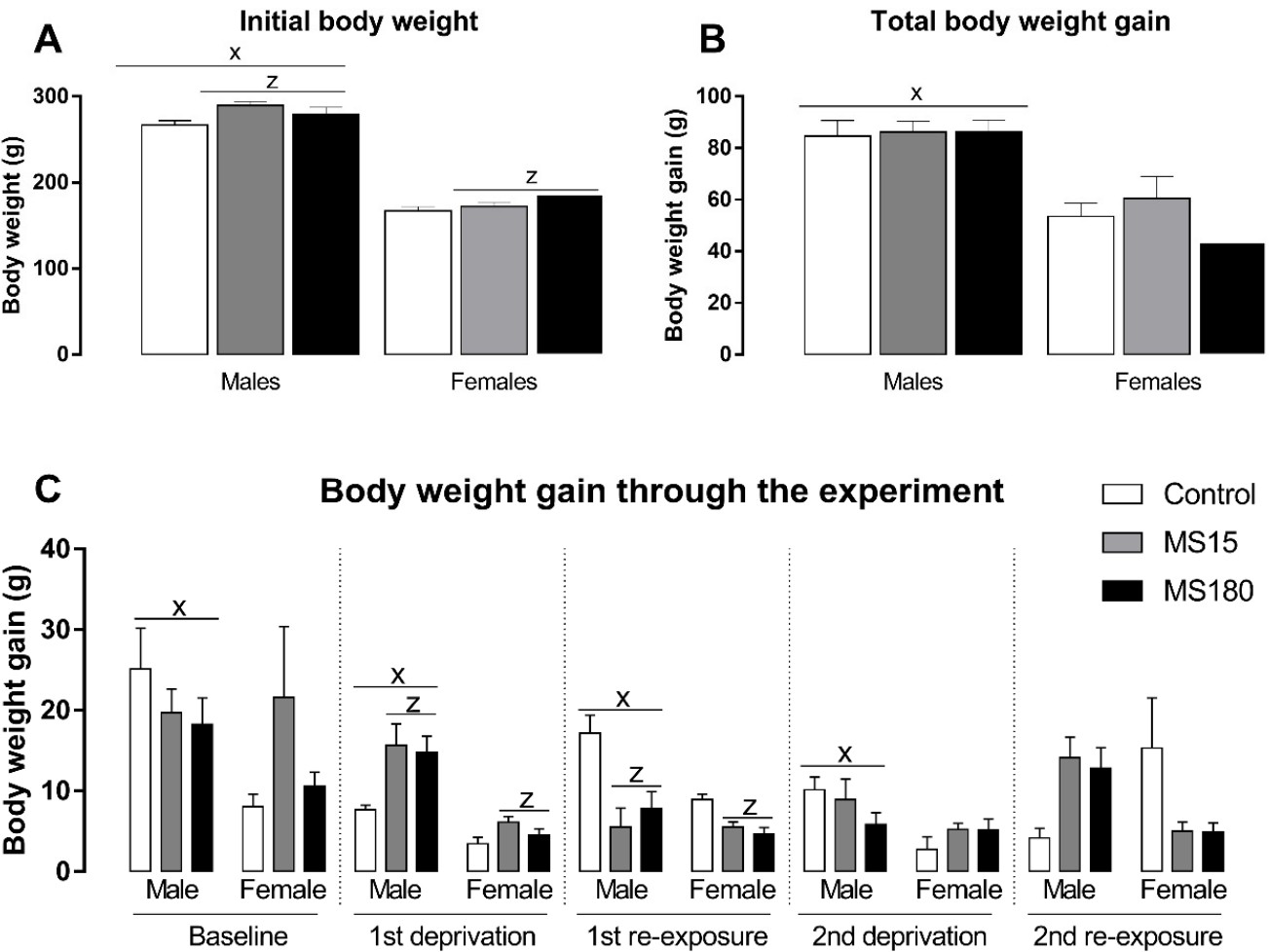

**Figure 5.** Initial body weight and body weight gain through the experiment in female and male rats exposed to brief and prolonged maternal separation. Bars represent the mean ± SEM (n = 11–14 animals/group). (**A**), initial body weight (g). (**B**), body weight gain during the whole period of alcohol deprivation effect (g/kg). (**C**), body weight gain in different phases of the alcohol deprivation effect protocol (g/kg). Body weight gain was calculated as the difference between the body weight on the last day of the respective phase and the first day of the same experimental phase. x; $p < 0.05$ when compared to female groups, independent of the manipulation; z, significant effect of maternal separation, regardless of sex. MS15, brief maternal separation; MS180, prolonged maternal separation.

## 4. Discussion

Here we showed that male and female rats are differentially affected by alcohol deprivation periods depending on the early-life experiences (brief or prolonged MS). First, females naturally drink more when compared to males. In males, the ADE was evident only after the 2nd ethanol re-exposure, and brief MS exposure was able to prevent it. In females, the ADE emerged earlier, in the 1st re-exposure period, while both brief and prolonged MS exposure prevented the ADE.

As already reported, we found remarkable differences in alcohol consumption between females and males. In both mice and rats, studies using several animal models of voluntary ethanol consumption showed that females consume more ethanol than males [16–20]. In a recent study from our lab, we showed that such differences are influenced by the concentration of the ethanol solution offered to the animals [16]. On the other hand, in humans, opposite results have been reported in epidemiological studies: men's ethanol intake and AUD prevalence are greater when compared to women, although women's

AUD prevalence has been growing recently [21]. Such results give strength to the view that alcohol drinking in humans is strongly influenced by cultural factors and that the underrepresentation of female individuals in biomedical research has a profound impact on substance use disorders research [41].

Our results also revealed that female Wistar rats are more susceptible to ADE since the increased alcohol intake in females showed up sooner than in males. As cited before, there is evidence suggesting that women who drink excessively are more prone to the development of medical problems compared to men [24], despite the greater incidence of AUD in men. In animal models, the results are highly dependent on the protocol used and the behavioral outcome tested. In this context, female rats showed enhanced sensitivity to the rewarding effects of ethanol [42] and increased ADE [19,43]. On the other hand, male rodents show increased behavioral signs of ethanol withdrawal [44] and greater withdrawal-induced ethanol-seeking behavior compared to females [45].

Studies in rodents, using mixed-sex analysis, showed that the prolonged disruption of pups-mother contact during early-life increases ethanol binge drinking and voluntary ethanol consumption in operant self-administration paradigms performed later in life [39,46–48]. Prolonged MS also increases ethanol consumption in male rats tested during adulthood in the 2-bottle [36,49,50] or 3-bottle choice paradigms [16]. On the other hand, brief MS during early life usually decreases ethanol intake in adult male rodents [36,50]. Both brief and prolonged periods of MS increase voluntary ethanol consumption in female rats submitted to a 3-bottle choice procedure [16], while there is no effect of MS when ethanol consumption is evaluated using the 2-bottle choice procedure [51,52]. Here, we found a more complex effect: while, in males, brief MS blocked the ADE, both brief and prolonged MS prevented the ADE in females. To date, few studies looked for the effects of MS on the ADE, especially using female individuals in the experimental procedure. One study found that neither brief nor prolonged MS affected the increased ethanol consumption after periods of intermittent ethanol exposure [53]. However, in this study, the authors did not use an animal facility hearing or handled control without separation periods, making difficult the comparison between the studies.

We found interesting results regarding body weight gain through different phases of our ADE protocol. As expected, females gained less body weight compared to males, regardless of the experimental phase, since male Wistar rats are larger and have an increased growth rate from puberty onwards than female Wistar rats [54]. Interestingly, animals exposed to MS, independent of MS length and sex, showed increased body weight. Moreover, maternally-separated rats showed increased body weight gain during periods of increased stress, that is, the first deprivation and first ethanol re-exposure phases. It should be related to an increased stress sensitivity in animals exposed to early-life stressors [35].

The variation in the period when brain maturation occurs could be related to the MS sex differences found in our study. Females' prefrontal cortex and amygdala undergo maturation faster when compared to males [13–15] and thus are more likely to be affected by the MS during the postnatal period that we performed our experiments. Prefrontal cortex dendritic morphology is more susceptible to the effects of MS in females than in males [55]. In this sense, the prefrontal cortex and the amygdala are involved in alcohol craving and relapse for alcohol drinking [56].

Two variables presented in our study could have impacted our results: the single-housing condition and the females' estrous cycle phase. First, social isolation is a known stressor for rodents and can affect voluntary ethanol intake [57,58]. This is an important limitation of our work since individual housing was necessary for the evaluation of ethanol intake. On the other hand, more recent works have been showing that social isolation does not impact consumption of ethanol solutions up to 10%, in both rats and mice [59,60]. Thus, the procedure used in our work should be taken into account when interpreting our results. Secondly, we did not evaluate the estrous cycle or synchronized females' cycle in our work. Literature reports have been showing that the estrous phase does not have a substantial impact on ethanol intake in rats [20], although it seems to influence

ethanol consumption microstructure [61]. It could be explained by the fact that ethanol administration itself disrupts the estrous cycle in rats [62,63]. Moreover, human studies are inconsistent regarding the correlation between the menstrual cycle phase and alcohol intake [64].

In conclusion, brief MS exerted a protective effect against the ADE in both female and male rats. Despite the increased susceptibility to the ADE shown by females, prolonged MS also prevented the ADE in females, revealing sex-dependent effects of MS in rats.

**Author Contributions:** M.C., G.M.-S. and G.A.B.C. performed the experiments. G.M.-S. analyzed the data and drafted the manuscript. M.T.M. and V.C.S.A. designed the study, supervised the project, and revised the manuscript for intellectual content. All authors have read and agreed to the published version of the manuscript.

**Funding:** This research was funded by Fundação de Amparo a Pesquisa do Estado de Goias (FAPEG/CNPq Programa Primeiros Projetos [201610267001023]).

**Institutional Review Board Statement:** Experiments were carried out according to the principles and standards of the Brazilian National Council for the Control of Animal Experimentation (CONCEA), based on NIH Guidelines for the Care and Use of Laboratory Animals as approved by the Commission on Ethics in Animal Use (CEUA) of the State University of Goias (protocol number 008/2016).

**Informed Consent Statement:** Not applicable.

**Data Availability Statement:** The data presented in this study are available on request from the corresponding author.

**Conflicts of Interest:** The authors declare no conflict of interest.

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
