# Peer review of "Alcohol Deprivation Differentially Changes Alcohol Intake in Female and Male Rats Depending on Early-Life Stressful Experience"

_neurosci, doi:10.3390/neurosci3020016_

Round 1
Reviewer 1 Report
Review of manuscript entilted: “Alcohol deprivation differentially changes alcohol intake in female and male rats depending on early-life stressful experience” authored by Marielly Carvalho, Gessynger Morais-Silva, Graziele Alícia Batista Caixeta, Marcelo Tadeu Marin and Vanessa Cristiane Santana Amaral
At the beginning I want to thank you for opportunity to review this interesting manuscript.
In the presented manuscript authors investigated sex differences in alcohol deprivation effect (ADE) in individuals affected by early-life stress caused by maternal separation. Introduction provides sufficient information about problem. Methods are described extensively and with details. Results are clearly presented in two tables and two figures. Discussion is easy to understand and written logically, based on obtained results.
Overall, manuscript is well-written and easy to follow. Authors obtained very interesting results, however due to simplicity of analyses (only ethanol and fluid intake are presented) it must be considered as a prelude to more advanced experiments. To sum up, I have only one remark.
Major concerns:
Minor concerns:
- I did not notice information whether females were synchronized in menstrual cycle or not. Different phases of menstrual cycle may have had impact on ethanol intake.
- I believe that ethanol consumed and fluid consumed tables could be converted into figures, to make it more reader friendly. However it is my personal point of view, which can be omitted.
Author Response
We would like to thank the reviewer for accepting to review our paper. Our point-by-point response to the reviewer’s comments could be found below, together with the initial reviewer's concerns. All the necessary alterations in the manuscript are highlighted.
- I did not notice information whether females were synchronized in menstrual cycle or not. Different phases of menstrual cycle may have had impact on ethanol intake.
Response: We agree to the reviewer’s point that this information is not clearly indicated in the manuscript. The estrous cycle was not evaluated and the female’s cycle was not pharmacologically synchronized in this work. We decided to not synchronize the cycle since reports in the literature have been showing that the estrous phase does not have a substantial impact on ethanol intake in rats (Priddy et al., 2017) and the synchronization treatments could per se impact the ethanol intake. Moreover, the length of the alcohol deprivation effect protocol made impractical the evaluation of the estrous cycle through the experiments. Additionally, we decided to expand our discussion with the implications of the estrous cycle to our results.
- I believe that ethanol consumed and fluid consumed tables could be converted into figures, to make it more reader friendly. However it is my personal point of view, which can be omitted.
Response: We agree with the reviewer’s point and changed the results accordingly.
References:
Priddy BM, Carmack SA, Thomas LC, Vendruscolo JCM, Koob GF, Vendruscolo LF (2017) Sex, strain, and estrous cycle influences on alcohol drinking in rats. Pharmacol Biochem Behav 152:61–67 Available at: http://dx.doi.org/10.1016/j.pbb.2016.08.001.
Reviewer 2 Report
In the current research, the authors performed experiments combining maternal separation and alcohol consumption and withdrawal paradigm. While potential findings can be interesting for readers several questions should be addressed and answered
1 My main point is obtained results are curious and should be discussed properly. It seems that short-term maternal separation is beneficial for males and even longer one is beneficial for females which is contradict with other known data. For example, maternal separation is an established model of depression phenotype. Authors should describe possible neurochemical or neurophysiological mechanisms behind it, and address that question in the introduction and discussion.
2 there is an increase in fluid and ethanol intake in females. There possible to calculate some kind of index of alcohol preference in different stages for males and females? it will be easier to distinguish just general increase in liquid consumption and preference specifically to alcohol
3 its not easy to read presented tables. May be present this data in figures, and make a table for alcohol preference?
4 Please indicate in the discussion, that single-house conditions can affect the result. It is known for previous studies that for example, social interaction and alcohol consumption both work via oxytocin. Social withdrawal can increase consumption, while social reunion can decrease alcohol consumption in rats /mice
Author Response
We would like to thank the reviewer for accepting to review our paper. Our point-by-point response to the reviewer’s comments could be found below, together with the initial reviewer's concerns. All the necessary alterations in the manuscript are highlighted.
1 My main point is obtained results are curious and should be discussed properly. It seems that short-term maternal separation is beneficial for males and even longer one is beneficial for females which is contradict with other known data. For example, maternal separation is an established model of depression phenotype. Authors should describe possible neurochemical or neurophysiological mechanisms behind it, and address that question in the introduction and discussion.
Response: We agree with the reviewer’s point and improved the discussion. Indeed, it was surprising to find that prolonged maternal separation blocked the alcohol deprivation effect in females. We hypothesize that the exposure to maternal separation impaired the normal maturation of the females’ brain to a greater extent than in males, especially in regions like the prefrontal cortex and amygdala. Since these regions are relevant to the alcohol deprivation effect and alcohol use disorders, such impairment could prevent the ethanol effects in those brain regions and consequently the expression of the alcohol deprivation effect. We extended the discussion to include this possible mechanism.
2 there is an increase in fluid and ethanol intake in females. There possible to calculate some kind of index of alcohol preference in different stages for males and females? it will be easier to distinguish just general increase in liquid consumption and preference specifically to alcohol
Response: We reported only ethanol intake (g/kg) since it is the most used data in studies involving the ADE. However, we agree with the reviewer’s point and decided to report the ethanol preference as well (figure 3C and D).
3 its not easy to read presented tables. May be present this data in figures, and make a table for alcohol preference?
Response: We agree with the reviewer’s point and changed the results accordingly. We transformed the ethanol intake prior deprivations and fluid intake tables into figures and added an alcohol preference figure as well (figure 3C and D).
4 Please indicate in the discussion, that single-house conditions can affect the result. It is known for previous studies that for example, social interaction and alcohol consumption both work via oxytocin. Social withdrawal can increase consumption, while social reunion can decrease alcohol consumption in rats /mice
Response: We agree with the reviewer’s point and indicated in the discussion section that single-housing should be taken into account when interpreting our results.
Reviewer 3 Report
The manuscript entitled „Alcohol deprivation differentially changes alcohol intake in female and male rats depending on early-life stressful experience“ by Carvalho and colleagues is about the long-lasting impact of short vs. long maternal separation on alcohol deprivation in male and female rats. The alcohol deprivation paradigm used is thought to model a relapse-like drinking situation. The authors found that a short but not a long maternal separation protected males from increased alcohol intake after a period of alcohol deprivation. Thus, the short maternal separation made the male rats more resilient to alcohol relapse. In females, both a short and a long maternal separation protected them from increased alcohol intake after alcohol deprivation
The manuscript is very interesting to read and provide new insights into sex differences in alcohol intake after a period of alcohol deprivation. However, I have a few points for the authors to address. Please see my comments below.
The Abstract is well written and clear. Please homogenize the past tense for obtained results.
The Introduction section is overall well written. A definition of the alcohol deprivation effect paradigm needs to be provided. Evidence for the relevance of this paradigm in rodents to ethanol addiction in humans also needs to be enclosed. Some background information are missing regarding the impact of maternal separation on addiction-relevant behaviour and the sex differences. Indeed, what is known about the impact of short vs. long maternal separation on alcohol self-administration? Motivation? Compulsive alcohol-seeking? What is known about the sex differences in these behaviours?
Minor comment: Some abbreviations needs to be defined before first mention (e.g. CPP line 679).
The Method section is very clear and contains all necessary experimental details. The experimental timeline is appreciated.
Minor comment: The authors need to justify the use of parametric statistical analyses (did the data meet the normal distribution and homogeneity of variances?).
In the Result section, the description of the effects of maternal separation on ethanol intake prior to deprivation (even if not statistically significant) are missing. Please avoid the adjective “big” to compare the ethanol intake, rather increased or decreased.
The reading of the results is very difficult due to the insertion of the details of the statistical analyses. For example, the authors write line 171: “Control males showed an increase in ethanol consumption only after the second deprivation (control/male baseline x control/male 1st deprivation, p = 0.8; control/male baseline x control/male 2nd deprivation, p < 0.05).” This can rewritten as “Compared to baseline, control males showed an increase in ethanol consumption after the second (p < 0.05) but not first deprivation period (p = 0.8).” This comment is true for all the text of the Result section.
Maybe the authors want to only mention the statistical outputs for the significant results, or compile them into a Table.
Minor comments: Why providing the exact p values for non-significant results and not for the significant ones? In the Table 1, 2 and Figure 2, is mean-SE referring to mean±SD or mean±SEM??
Did the authors monitor the body weight of the animals as well as the food intake? Did the authors check for behavioural signs of ethanol withdrawal?
The Discussion section is overall well written. I would only suggest the authors to discuss more the possible neurobiological mechanisms by which i) females are more vulnerable to alcohol drinking than males, ii) maternal separation, regardless of the length, protects females but not males from alcohol relapse, iii) a short but not long maternal separation protects males from alcohol relapse.
Author Response
We would like to thank the reviewer for accepting to review our paper. Our point-by-point response to the reviewer’s comments could be found below, together with the initial reviewer's concerns. All the necessary alterations in the manuscript are highlighted.
The Abstract is well written and clear. Please homogenize the past tense for obtained results.
Response: We would like to thank the reviewer for accepting to review our paper. The abstract was checked to ensure that results are described in the past tense.
The Introduction section is overall well written. A definition of the alcohol deprivation effect paradigm needs to be provided. Evidence for the relevance of this paradigm in rodents to ethanol addiction in humans also needs to be enclosed. Some background information are missing regarding the impact of maternal separation on addiction-relevant behaviour and the sex differences. Indeed, what is known about the impact of short vs. long maternal separation on alcohol self-administration? Motivation? Compulsive alcohol-seeking? What is known about the sex differences in these behaviours?
Response: We would like to thank the reviewer for the comments and agree that the introduction section was missing such information and so we improved the text accordingly. Moreover, some of the requested background information is scarce, since few studies effectively compared female and male rats using protocols of maternal separation and alcohol addiction-related behaviors. We hope that the new information added to the introduction and the data used in the discussion should be enough to surpass such weakness of the manuscript.
Minor comment: Some abbreviations needs to be defined before first mention (e.g. CPP line 679).
Response: The text was revised to correct such mistakes.
The Method section is very clear and contains all necessary experimental details. The experimental timeline is appreciated.
Response: We would like to thank the reviewer for the enriching comments.
Minor comment: The authors need to justify the use of parametric statistical analyses (did the data meet the normal distribution and homogeneity of variances?).
Response: We used repeated-measures ANOVA to analyze our data since it is a robust test to detect significant differences in study designs that uses continuous variables that are repeated obtained over time (Yan et al., 2017). This test is largely employed in studies with similar designs to ours (for example, see Vengeliene et al., 2014). Moreover, in general, non-parametric tests lack power when compared to their parametric counterparts (Whitley and Ball, 2002; Fay and Gerow, 2013). For a small number of samples, repeated measures ANOVA is preferred over Multi-Level Linear Models (Haverkamp and Beauducel, 2017) relative to type-I error. Thus, when designing our experiments, we decided to use parametric statistical analyses.
In the Result section, the description of the effects of maternal separation on ethanol intake prior to deprivation (even if not statistically significant) are missing. Please avoid the adjective “big” to compare the ethanol intake, rather increased or decreased.
Response: The results section was revised to correct such mistakes.
The reading of the results is very difficult due to the insertion of the details of the statistical analyses. For example, the authors write line 171: “Control males showed an increase in ethanol consumption only after the second deprivation (control/male baseline x control/male 1st deprivation, p = 0.8; control/male baseline x control/male 2nd deprivation, p < 0.05).” This can rewritten as “Compared to baseline, control males showed an increase in ethanol consumption after the second (p < 0.05) but not first deprivation period (p = 0.8).” This comment is true for all the text of the Result section.
Response: We edited the results section to make the text more reader-friendly.
Maybe the authors want to only mention the statistical outputs for the significant results, or compile them into a Table.
Response: We decided to mention only the significant results of the statistical analysis to keep the text more reader-friendly.
Minor comments: Why providing the exact p values for non-significant results and not for the significant ones? In the Table 1, 2 and Figure 2, is mean-SE referring to mean±SD or mean±SEM??
Response: There is no specific reason for such discrepancy. We decided to use only approximated values for both non-significant and significant results. Regarding mean±SE, we were referring to mean±SEM. We changed the text accordingly.
Did the authors monitor the body weight of the animals as well as the food intake? Did the authors check for behavioural signs of ethanol withdrawal
Response: We evaluated body weight from the beginning of the alcohol deprivation effect protocol. The repeated-measures ANOVA of the body weight obtained in all experimental days (from the first day of ethanol from the last day of the second re-exposure) showed significant effects for maternal separation (F2,69 = 6.08; p < 0.01), sex (F1,69 = 664.55; p < 0.001), time (F35,2415 = 322.83; p < 0.001), for the interaction between sex and time (F35,2415 = 23.81; p < 0.001) and, for the interaction between maternal separation, sex and time (F70,2415 = 1.87; p < 0.001). Due to the big (and interesting) number of differences found between groups (data not shown in the manuscript), we decided to split the analyses to include them in the manuscript. First, we found a significant effect in initial body weight for the factor maternal separation (F2,69 = 5.06; p < 0,01) and sex (F1,69 = 567.40; p < 0.001). The posthoc analysis revealed that males began the alcohol deprivation effect protocol heavier than females (p < 0.001), and both brief or prolonged maternal separation increased body weight in both males and females (p < 0.05). On the other hand, total body weight through the experiment was only affected by sex (F1,69 = 55.57; p < 0.001), revealing that males gain more weight than females over time (p < 0.001). Next, we evaluated the body weight gain in each phase of our protocol. Body weight gain during the baseline phase was affected only by sex (F1,69 = 4.21; p < 0.05), revealing a greater body weight gain in males compared to females (p < 0.05). During the first deprivation period, the two-way ANOVA revealed a significant effect for maternal separation (F2,69 = 5.51; p < 0,01) and sex (F1,69 = 35.44; p < 0.001), showing that males gained more weight during this phase (p < 0.001) and maternal separation increased body weight gain during the ethanol deprivation phase (p < 0.05). We found similar results for the first re-exposure phase: a significant effect for maternal separation (F2,69 = 10.88; p < 0,01) and sex (F1,69 = 7.19; p < 0.001), showing that males gained more weight during this phase (p < 0.001) and maternal separation increased body weight gain during the ethanol deprivation phase (p < 0.001). We found a significant effect only for sex during the second deprivation phase (F1,69 = 7.90; p < 0.01), when males gained more weight than females (p < 0.01). Finally, we found a significant effect for the interaction between maternal separation and sex (F2,69 = 7.63; p < 0.01) in the second re-exposure phase, while the posthoc test did not reveal any significant differences. We included the body weight results in the results section of the manuscript, depicted in figure 5. Unfortunately, food intake and ethanol withdrawal were not monitored in our work.
The Discussion section is overall well written. I would only suggest the authors to discuss more the possible neurobiological mechanisms by which i) females are more vulnerable to alcohol drinking than males, ii) maternal separation, regardless of the length, protects females but not males from alcohol relapse, iii) a short but not long maternal separation protects males from alcohol relapse.
Response: We agree with the reviewer’s point and improved the discussion. Indeed, it was surprising to find that prolonged maternal separation blocked the alcohol deprivation effect in females. We hypothesize that the exposure to maternal separation impaired the normal maturation of the females’ brain to a greater extent than in males, especially in regions like the prefrontal cortex and amygdala. Since these regions are relevant to the alcohol deprivation effect and alcohol use disorders, such impairment could prevent the ethanol effects in those brain regions and consequently the expression of the alcohol deprivation effect. We extended the discussion to include this possible mechanism.
References:
Fay DS, Gerow K (2013) A biologist’s guide to statistical thinking and analysis. WormBook:1–54.
Haverkamp N, Beauducel A (2017) Violation of the sphericity assumption and its effect on type-I error rates in repeated measures ANOVA and multi-level linear models (MLM). Front Psychol 8:1–12.
Vengeliene V, Bilbao A, Spanagel R (2014) The alcohol deprivation effect model for studying relapse behavior: A comparison between rats and mice. Alcohol 48:313–320 Available at: http://dx.doi.org/10.1016/j.alcohol.2014.03.002.
Whitley E, Ball J (2002) Statistics review 6: Nonparametric methods. Crit Care 6:509–513.
Yan F, Robert M, Li Y (2017) Statistical methods and common problems in medical or biomedical science research. Int J Physiol Pathophysiol Pharmacol 9:157–163.
Round 2
Reviewer 2 Report
The authors did a good job and respond to all questions and modified the manuscript accordingly. So I recommend accepting this paper for publication.